# Use of CGF in Oral and Implant Surgery: From Laboratory Evidence to Clinical Evaluation

**DOI:** 10.3390/ijms232315164

**Published:** 2022-12-02

**Authors:** Andrea Palermo, Laura Giannotti, Benedetta Di Chiara Stanca, Franco Ferrante, Antonio Gnoni, Paola Nitti, Nadia Calabriso, Christian Demitri, Fabrizio Damiano, Tiziano Batani, Massimo Lungherini, Maria Annunziata Carluccio, Biagio Rapone, Erda Qorri, Antonio Scarano, Luisa Siculella, Eleonora Stanca, Alessio Rochira

**Affiliations:** 1College of Medicine and Dentistry Birmingham, University of Birmingham, Birmingham B4 6BN, UK; 2Department of Biological and Environmental Sciences and Technologies, University of Salento, 73100 Lecce, Italy; 3Independent Researcher, 73100 Lecce, Italy; 4Department of Basic Medical Sciences, Neurosciences and Sense Organs, University of Bari Aldo Moro, 70121 Bari, Italy; 5Department of Engineering for Innovation, Campus Ecotekne, University of Salento, Via per Monteroni, 73100 Lecce, Italy; 6National Research Council (CNR) Institute of Clinical Physiology (IFC), 73100 Lecce, Italy; 7Silfradent srl, via A. Moro, 22, 73100 Lecce, Italy; 8Interdisciplinary Department of Medicine, Aldo Moro University of Bari, 70121 Bari, Italy; 9Faculty of Medical Science, Albanian University, Bulevardi Zogu I, 1001 Tirana, Albania; 10Department of Oral Science, Nano and Biotechnology and CeSi-Met, University of Chieti-Pescara, 66100 Chieti, Italy

**Keywords:** CGF, growth factor, stem cells, blood-derived biomaterials, osteogenic differentiation, dental implants, dental implantology, osseointegration

## Abstract

Edentulism is the condition of having lost natural teeth, and has serious social, psychological, and emotional consequences. The need for implant services in edentulous patients has dramatically increased during the last decades. In this study, the effects of concentrated growth factor (CGF), an autologous blood-derived biomaterial, in improving the process of osseointegration of dental implants have been evaluated. Here, permeation of dental implants with CGF has been obtained by using a Round up device. These CGF-coated dental implants retained a complex internal structure capable of releasing growth factors (VEGF, TGF-β1, and BMP-2) and matrix metalloproteinases (MMP-2 and MMP-9) over time. The CGF-permeated implants induced the osteogenic differentiation of human bone marrow stem cells (hBMSC) as confirmed by matrix mineralization and the expression of osteogenic differentiation markers. Moreover, CGF provided dental implants with a biocompatible and biologically active surface that significantly improved adhesion of endothelial cells on CGF-coated implants compared to control implants (without CGF). Finally, data obtained from surgical interventions with CGF-permeated dental implants presented better results in terms of optimal osseointegration and reduced post-surgical complications. These data, taken together, highlight new and interesting perspectives in the use of CGF in the dental implantology field to improve osseointegration and promote the healing process.

## 1. Introduction

Partial or total edentulism could be replaced with removable dentures, and such a solution is often rejected by the patient due to poor functionality and stability. The implant-retained and implant-supported prosthesis have become a common device that improves the treatment of edentulous patients [1,2,3,4]. Osseointegration is a critical and fundamental process for the favorable outcome of the dental implant. The most important aspects for successful osseointegration are the biological characteristics of the host site (the patient) and the macro- and micro-structure of the prosthetic implant [5,6].

Dental implant surfaces have now achieved outstanding performances and several physical, mechanical, or chemical procedures have been used to increase roughness, resulting in improved wettability [7]. Common implant surfaces are classified into two great categories: smooth and treated. It has been clearly demonstrated that micro-rough surface implants determine a greater and faster new bone tissue addition when compared to smooth surface implants.

Osteoblast differentiation and its implication in the osseointegration process are affected by implant surface nano- and micro-topographical characteristics. Several phenomena, such as clot formation, fibrin texture retention, and cell population differentiation, are influenced by the surface topography. The biomechanical processes occurring at the interface between the tissues and the implant surface, as well as the implant–prosthetic material are governed by the implant design (external micro- and macro-structure), the patient response, the surgical technique, by the conditions, and by the loading times. Air exposure of the implant surface leads to an oxidized layer formation which represents a suitable substrate for its interaction with body fluids, the first and fundamental mediator of all biological processes. Implant insertion and the consequent surgical trauma cause bone blood vessel interruption, with subsequent bleeding: this determines contact between biological fluids of the host and the inserted implant surface. The absorption of ions and macromolecules of blood origin on the implant surface is immediate and important both for the platelet adhesion itself and for the consequent osteogenesis. Implant surfaces have achieved remarkable performance, thus guaranteeing highly successful osseointegration [8].

However, the success of dental implants and optimal control in the post-operative period in elderly subjects and in patients with chronic degenerative diseases is still under debate; therefore, new strategies are needed to accelerate wound healing and prevent or mitigate post-operative complications even in unfavorable conditions. Further improvements to modern implants can be obtained biologically by adding autologous growth factors, obtained by processing the patient’s venous blood.

Based on their characteristics and methods of preparation, platelet derivatives can be classified into three different generations. The first generation is platelet-rich plasma (PRP), which contains several growth factors involved in tissue repair, but the use of anticoagulants and bovine thrombin is required to induce fibrin polymerization [9,10]. Currently, the PRP has been divided into two subclasses: pure platelet-rich plasma (P-PRP) and leukocyte- and platelet-rich plasma (L-PRP) [9].

The second generation consists of platelet-rich fibrin (PRF) and can be classified into three subgroups: pure platelet-rich fibrin (P-PRF), leukocyte- and platelet-rich fibrin (L-PRF), and injectable PRF (I-PRF) [9]. For its preparation, blood samples are collected without the use of anticoagulants or biological agents [10].

The third and latest generation of platelet concentrate products is concentrated growth factors (CGF), developed by Sacco in 2006 [11]. CGF is produced by the centrifugation of venous blood without the addition of any exogenous product and is therefore free from cross-contamination. Repeated switch of the centrifugation speed leads to the production of CGF with high amounts of cytokines, platelets, nucleated cells, and very dense fibrin scaffolds [12,13,14,15]. CGF, as a platelet concentrate, is rich in multiple growth factors released by activated platelets capable of mediating inflammation, angiogenesis, and wound repair [16,17,18]. The autologous growth factors can be released by CGF gradually over a period of time, playing a crucial role in hard and soft tissue repair [17,18]. Moreover, the fibrin scaffold of CGF, or products obtained from its processing, entangle stem cells able to differentiate into several cell types [17,18,19,20,21]. Cell-based therapy for the regeneration of bone tissue has been extensively investigated [22,23]. Recently, we demonstrated that CGF induced osteogenic differentiation of human bone marrow stem cells (hBMSC) and promoted endothelial angiogenesis due to the release of soluble and cellular components suggesting CGF as a biomaterial for therapeutic vasculogenesis in the field of tissue regeneration. [10,18].

However, the potential role of the CGF in improving the process of osseointegration of dental implants is unclear. In the present study, we used an innovative method which allowed us to permeate the titanium dental implants with CGF before their implantation in the patients. The aim of this work was to investigate the effects of CGF-permeated dental implants on the osseointegration process both in in vitro and in vivo models.

## 2. Results

### 2.1. Biological Characterization of the CGF-Permeated Implants

To achieve a biologically active surface, we permeated IML dental implants with CGF, using the Round up device (Silfradent).

To verify the interaction between CGF and the titanium implant surface, we performed SEM analysis (Figure 1). The micrographs revealed that the Round up device allowed permeation of the surface of implants with CGF almost completely; the upper end of the implant was the only one uncovered (Figure 1A). CGF fibrin covered the implant forming a dense network (Figure 1B,C). Furthermore, the implant surface showed some corpuscular elements entangled in the fibrin network (Figure 1D), as observed for the CGF alone [17].

To characterize the biological properties of CGF-permeated implants, we analyzed the release of crucial soluble factors in wound healing and tissue regeneration including growth factors and matrix metalloproteinases (MMPs). Among the growth factors, we evaluated the appearance of vascular endothelial growth factor (VEGF), transforming growth factors β1 (TGF-β1), and bone morphogenetic protein-2 (BMP-2). We also quantified MMP-2 and MMP-9 matrix-degrading enzymes involved in cell migration and tissue remodeling.

In order to mimic the natural release of bioactive molecules from the CGF-coated implants, we incubated the CGF-permeated implants in culture medium for 0–28 days. After each incubation period (1, 7, 21, and 28 days), we collected aliquots of medium conditioned by CGF-coated implants (I-CGF) and we analyzed the soluble factors released in the culture medium. We found a release of growth factors, such as VEGF, TGF-β1, and BMP-2, at each experimental time, after CGF-permeated implant incubation. As reported in Figure 2A, VEGF was detected in I-CGF after 1 day from the preparation of the permeated implants, reached a maximum value after 7 days (107 ng/mL), and gradually decreased over time until it reached a value of 42 ng/mL after 28 days.

TGF-β1 and BMP-2 appeared to be released slowly, reaching a high point after 21 days and remaining high for up to 28 days. The TGF-β1 content was about 2.6 ng/mL, 7.4 ng/mL, and 6.9 ng/mL after 1, 21, and 28 days, respectively (Figure 2B). The amount of BMP-2 was about 1.8 pg/mL after one day, 6.7 pg/mL after 21 days, and 5.9 pg/mL after 28 days (Figure 2C).

The release of matrix metalloproteinases MMP-9 and MMP-2 by CGF-coated implants was also evaluated. The release kinetics of MMP-9 (Figure 2D) and MMP-2 (Figure 2E) highlighted a peak on the first day and gradually decreased over time. Indeed, one day after the preparation of CGF-permeated implants, MMP-2 and MMP-9 in I-CGF were about 5.1 ng/mL and 86.2 ng/mL, respectively, and significantly lowered after 28 days (1.2 ng/mL ng for MMP-2 and 6.1 ng/mL ng for MMP-9) (Figure 3D,E).

In order to clarify whether the CGF produces a biologically active surface around dental implants able to favor the adhesion of surrounding cells, we evaluated the adhesion on the surface of the implants by human endothelial cells, which play a crucial role in tissue repair and regeneration. For this purpose, human cultured microvascular endothelial cells, HMEC-1, were labeled with vital fluorescent dye DilC18, and seeded on the implants. After 16 h, we monitored by fluorescence microscopy the HMEC-1 adhesion on implants permeated with CGF and compared with control implants (without CGF).

Figure 3 shows that the CGF-permeated implants had a biocompatible and biologically active surface, which was able to improve the adhesion of endothelial cells on CGF-coated implants compared to the implants without CGF. Our findings confirm CGF as a treasured biomaterial and suggest a role for the CGF-coating of dental implants in improving the process of tissue regeneration.

### 2.2. Osteogenic Properties of the CGF-Permeated Implants

To further assess the effects of CGF-permeated implants on osteogenic differentiation, the matrix mineralization of BMSC was evaluated by alizarin red staining (ARS). As expected, the osteogenic medium caused the formation of mineralized nodules in hBMSC in vitro after 21 days, which was not evident in untreated control hBMSC (Figure 4). In hBMSC, the matrix mineralization requires the addition of substrates as β-glycerophosphate (BGP) and ascorbic acid 2-phosphate (AA) [10,24,25,26]. Consequently, cells treated with the CGF-permeated implant were incubated with BM plus BGP and AA. In these conditions, ARS assays showed strong BMSC mineralization demonstrating a positive effect of the CGF-permeated implant on the osteogenic differentiation process (Figure 4).

To verify the BMSC osteogenic differentiation induced by the CGF-permeated implant, the expression of some molecular targets, typically activated during osteogenic differentiation, was investigated. The mRNA abundance of the transcription factor RUNX2, a key regulator of osteogenesis, the extracellular matrix proteins COL1a1 and OCN, was quantified (Figure 5). The cells incubated with the CGF-permeated implant showed a significant increment of RUNX2 and OCN mRNA with respect to the control; indeed, the expression of RUNX2 and OCN increased by about 70% and 124%, respectively. RUNX2 and OCN mRNA levels significantly increased in osteogenic-medium-treated cells, with respect to the control, by about 46% and 152%, respectively, as expected. COL1a1 expression behaved differently; the osteogenic medium and CGF-permeated implant treatments induced a statistically significant reduction of COL1a1 mRNA abundance when compared to control by about 75% and 50%, respectively (Figure 5).

### 2.3. Clinical Evaluation of CGF-Permeated Implants

To validate the performance of CGF-permeated dental implants in vivo, we recruited 10 healthy subjects, aged between 60 and 72, to be rehabilitated with 20 implants of which 10 were CGF-permeated. The split-mouth design, which provides rehabilitating one half arch with traditional implants, and the other half arch with CGF-permeated implants in the same patient, was used.

The post-operative course was followed over time and evaluated for complications such as pain and on probing bleeding and osseointegration was verified.

Pain was measured according to the VAS scale as previously described [27].

The pain experienced on the first day by patients of the CGF-permeated implant group was significantly less in comparison to the traditional implant group and was similarly reduced 7 days after surgery (Figure 6).

Moreover, the use of CGF-permeated implants showed an improvement in post-surgical complications such as bleeding on probing at six months, compared to traditional implants, although the results did not reach statistical significance, due to the limited number of subjects (Table 1).

Noteworthy, those subjects showed an improved successful implantation on the rehabilitated side with CGF-permeated implants in terms of faster and easier healing and better long-term implant outcome.

Intraoral radiography showed that the CGF-permeated implants had no significant resorptions. Figure 7 shows some exemplary images of dental implants with or without CGF permeation performed in healthy subjects. Note that, in healthy subjects, CGF-permeated implants have a better outcome in terms of pain, swelling, and bleeding, seen at 24 h, and better long-term implant outcome, as seen at 6 months after surgery.

Radiographic images reporting results of implants executed with or without CGF-permeated devices are reported in Figure 7. A better crestal level and probing were obtained using the CGF-permeated implants (Figure 7A, IV quadrant (a); Figure 7B, I quadrant (a)), compared to bone resorption observed with the un-permeated implant (Figure 7A, III quadrant (b); Figure 7B, II quadrant (b)).

## 3. Discussion

In recent years, CGF was widely recognized as an autologous blood derivative able to promote tissue repair affecting vascularization, cell migration, and differentiation [10,17,18,28,29,30]. Tissue repair is a complex mechanism that takes place through inflammatory processes, cell proliferation, differentiation, and matrix remodeling. Several mediators can be involved including cytokines, growth factors, and matrix-degrading enzymes [31]. Despite the large literature on CGF use and applications in the regenerative medicine field [30,32], up to the present, few data are provided on the effects of CGF in improving the bioactivity of the dental implant surface as well as osseointegration processes and tissue regeneration.

The osseointegration is defined as a direct structural and functional connection between the living bone and the surface of the load-carrying implant. This process is critical for implant stability and is considered a prerequisite for implant loading and long-term clinical success of osseous dental implants [33]. The key steps in osseointegration are the tissue response to the implantation, and the peri-implant osteogenesis and bone remodeling. Implant design and composition, patient systemic factors, surgical technique, and loading characteristics can all affect the success of osseointegration [34].

Dental implant surfaces have now achieved outstanding performances. Common implant surfaces are classified into two great categories: smooth and treated. The implant micro-surface and nano-surface modifications have been proven to affect cellular responses such as cell adhesion, proliferation, differentiation, and migration, thus influencing bone healing. Due to surface modifications, it was possible to overcome the adverse effects of length reduction and the unfavorable crown–implant ratio of short implants, shortening the time needed to achieve secondary stability and deliver prosthetic restoration [35]. The decontamination of the implant surface is also of fundamental importance. This procedure is useful for prosthetic loading and to minimize peri-implantitis, which is the main cause of implant failure [36].

In the present study, we used an innovative method capable of permeating titanium dental implants with CGF in order to make the surface of the device biologically active, before their implantation in patients. The aim of this work was to investigate the effects of CGF-permeated titanium dental implants on the osseointegration process by analyzing the molecular mechanisms involved.

Results reported in this study showed that dental implants were efficiently permeated with CGF. The CGF-permeated implants presented a dense network of fibrin with trapped corpuscular elements. These observations agree with our previous studies reporting the presence of resident cells with different morphology into the CGF scaffold [17,18]. We reported that CGF entangled, in the fibrin matrix, a large number of cells with stem cell features that regulated the production and sustained the release of CGF-soluble mediators, including growth factors and MMPs.

In order to verify whether the CGF layer covering the implants retained the ability to release over time soluble mediators involved in wound repair and tissue regeneration, as reported earlier for the CGF scaffold [17], in this work we incubated the CGF-permeated implants in the culture medium and analyzed the presence of growth factors and MMPs.

We found that CGF-permeated implants released into the surrounding environment over time both growth factors, including VEGF, TGF-β1, and BMP-2, and matrix-degrading enzymes, such as MMP-2 and MMP-9.

Among the factors released by the CGF, VEGF is a crucial molecule in tissue repair and regeneration being implicated in post-natal neo-vascularization in terms of angiogenesis, the formation of new capillaries from pre-existing vessels by mature endothelial cells, and vasculogenesis, the de novo vessel growth by bone-marrow-derived endothelial progenitor cells [37,38]. Due to the very short half-life of VEGF [39], a system of prolonged release of this growth factor is necessary to provide effective therapeutic action. This challenge can be overcome by the use of CGF, which could guarantee a prolonged release of VEGF over time.

In addition to VEGF, the implant permeated with CGF also released the growth factor TGF-β1, which is critical at wound healing sites and particularly in the oral cavity, where different types of cells, such as fibroblasts and osteoblasts, must be stimulated to proliferate and promote the process of tissue regeneration/remodeling [40].

BMP-2, another important member of the TGF-β superfamily, plays a prominent role in bone and cartilage development. It is a growth factor that can promote the differentiation and maturation of osteoblasts [41]. 

Our findings, reporting a sustained release of BMP-2 over time from CGF-permeated implants, suggest that the coverage of devices with CGF could improve the repair processes at the injury site. In addition to growth factors, we also found that CGF released matrix-degrading enzymes involved in many biological processes, including inflammation and cell migration during wound healing and tissue repair in coordination with different growth factors and cytokines [42].

Moreover, our results demonstrated that CGF-permeated implants possessed a biocompatible and biologically active surface which improved the adhesion of endothelial cells on CGF-coated implants with respect to the untreated traditional implants.

The fibrin matrix, covering dental implants, appears as an excellent substrate for the adhesion, migration, and invasion of endothelial cells, and subsequent formation of new capillary-like structures [19,43]. Endothelial cells can be recruited from tissues adjacent to the wound, bind to fibrin through various cell adhesion receptors, and lead to the formation of new vascular structures [18,44]. In addition to fibrin, growth factors and MMPs, concentrated in the CGF and gradually released, help to promote the adhesion, proliferation, migration of endothelial cells, and new capillaries formation, favoring the integration of CGF-permeated implants.

The results here reported showed that the CGF provided the implants with a biologically active and biocompatible surface for endothelial cells, improving their adhesion, which is a fundamental step for inducing angiogenesis and wound healing with promoting effects on the stability of the implants and confirming CGF as a valuable biomaterial to be applied also in the field of dental implants.

Recently, we demonstrated the ability of CGF to release primary cells, capable of differentiating into osteoblasts by producing a mineralized matrix [17], and to promote the osteogenic differentiation of stem cells [10]. Here we found that CGF permeated on the implants was able to induce hBMSC differentiation in osteoblasts, as supported by the evaluation of hBMSC matrix mineralization. The hBMSC osteogenic differentiation induced by the CGF-permeated implants was supported by the increased expression of the transcription factor RUNX2, key regulator of osteogenesis, and of two extracellular matrix proteins COL1a1 and OCN. It is interesting that a similar trend of osteogenic marker gene expression has been previously observed using the whole CGF [10]. Several studies supported the role of the BMP-2-triggered signaling pathway in mediating the CGF induction of mesenchymal cells’ osteogenic differentiation, through the stimulation of RUNX2 and OCN expression [29,45,46]. It has also been reported that CGF stimulated the proliferation and osteogenic differentiation of gingiva-derived mesenchymal stem cells by regulating the expression of BMP2 and RUNX2 [46], and upregulating the expression and secretion of osteogenic differentiation markers, including COL1 and OCN, in rabbit periosteum-derived cells [29]. Thus, the present findings suggested that the continuous and prolonged release of multiple bioactive factors over time by the CGF-coated implants can stimulate the complex and long process of tissue regeneration.

These findings were achieved in preclinical models, known as reliable tools for studying osteogenesis and the endothelial biocompatibility of CGF-permeated devices. However, our in vitro experimental results do not necessarily translate directly into the in vivo situation. Therefore, the following investigations have been performed to evaluate the in vivo efficacy of CGF-permeated implants in promoting osseointegration and tissue regeneration, and reducing post-surgical complications.

Data obtained from surgical interventions with CGF-coated dental implants presented better results in terms of osseointegration as well as of post-surgical complications. In particular, subjects enrolled in the present study reported that CGF-permeated implants produced a lower level of pain than traditional implants 1 day after surgery. The effect of CGF in lowering the level of pain is in agreement with another previous study by Taschieri et al. [47] showing the positive influence of CGF on quality of life by minimizing post-operative discomfort after dental implant rehabilitation. Furthermore, CGF was found to be effective in reducing pain levels after extraction of the mandibular third molar [27].

We also reported data regarding bleeding on probing and dental implant success 6 months after dental implant surgery. We found that CGF improved the success of the dental implant and the post-surgical complications, observing a trend in the reduction of bleeding on probing of CGF-permeated implants compared to traditional implants. Our results agree with a recent study by Shetye et al. [48] reporting that CGF accelerated osseointegration and had a positive effect on stabilization values.

Nevertheless, the results regarding the beneficial effects of CGF in dental implants are still conflicting. Özveri Koyuncu et al. [49] found that the use of CGF during dental implant surgery had a neutral effect on implant stability compared to traditional implants. However, in the studies conducted so far, the CGF was inserted into the dental socket prior to the insertion of the dental implant.

Otherwise, we obtained an implant permeated with CGF before its implantation in the dental cavity. This was an innovative, easy, and fast method based on the permeation of titanium dental implants with CGF, obtaining a biologically active surface ready to be used in dental rehabilitation. Here we characterized the biological properties of these implants in terms of the release of key factors in wound healing and tissue regeneration including growth factors and matrix metalloproteinases. Furthermore, we reported the osseointegration properties of CGF-permeated implants and verified the effects in vivo.

We had excellent results in all patients treated with both immediate loading and conventional (late) loading. Immediate loading provides benefits such as short treatment time, the elimination of the second surgery required for later loading protocols, the protection of the gingival papilla, an immediate esthetic effect, and high patient satisfaction [50]. However, no significant differences in the survival rate were reported between the immediately and conventionally loaded implants.

The importance of correct implant rehabilitation is also fundamental for improving one’s real and perceived quality of life. Numerous studies show that this phenomenon is particularly evident in vulnerable groups such as the elderly [51]. Most of the older adults perceived that the most affected dimensions were psychological discomfort and functional limitation. Age, educational level, marital status, type of insurance, and level of income had a statistically significant relationship with oral-health-related quality of life.

Overall, our findings highlighted CGF as a healing biomaterial that can be utilized in dental implantology surgical procedures to fasten healing and reduce post-operative discomfort, with positive effects in the success of the dental implant and suggested new interesting perspectives in the use of CGF in tissue repair and regeneration.

## 4. Materials and Methods

### 4.1. Preparation of CGF-Coated Dental Implants

For the in vitro analysis, blood samples of 8 mL were taken via venipuncture from ten non-smokers and in good general health donors, of which there were three females and seven males, aged between 60 and 72. Informed consents were obtained from the donors included in this study in accordance with the Declaration of Helsinki. Tubes of blood were processed to obtain a CGF clot, by a centrifuge (Medifuge MF200; Silfradent SrL, Forlì, Italy), using a program with the following characteristics: 30 s acceleration, 2 min 2700 rpm, 4 min 2400 rpm, 4 min 2700 rpm, 3 min 3000 rpm, and 36 s deceleration and stop, as previously described [10]. The resulting CGF was then inserted into dedicated tubes each containing an IML implant (Immediateload ^®^, Swiss dental implants, diameter 4 mm, height 8 mm), so that the coating procedure could be conducted in a closed field. To incorporate the CGF onto the implant surface, the tubes containing CGF and implant (Figure 8) were inserted into a second device, Round up (Silfradent srl, Forlì, Italy), and centrifuged for 16 s following the manufacturer’s instructions.

### 4.2. Scanning Electron Microscopy Analysis

The implants coated with CGF were fixed in 4% (*w*/*v*) paraformaldehyde (PFA) in PBS for 2 h (room temperature), followed by two PBS washings and a final storage in 0.05% (*w*/*v*) sodium azide in PBS. After fixation, they were rinsed twice with PBS and dehydrated in scalar ethanol/water solutions (15%, 25%, 50%, 70%, 90%, and 100% ethanol, 10 min each), then freeze-dried and coated with a 7 nm layer of gold and examined under scanning electron microscopy (SEM EVO^®^ 40, Carl Zeiss AG, Oberkochen, Germany), in a variable pressure mode and with an accelerating voltage of 20 kV. The samples were placed on the SEM sample holder, using double-sided adhesive tape, and were observed without any further manipulation, at different magnifications (150X, 1.00KX, 15.00KX, and 40.42KX). SEM micrographs were then analyzed by ImageJ 1.50c software (NIH, http://rsb.info.nih.gov/ij, accessed on 18 March 2021), in order to evaluate the average fiber diameter and the size distribution in the fibrin matrix (50 measurements for each acquired sample).

### 4.3. Growth Factors Release

CGF-permeated implants were placed in a 24-well plate (one in each well) with the addition of 1 mL of cell culture medium (L-DMEM), supplemented with 100 U/mL penicillin/streptomycin and without fetal bovine serum (FBS), and incubated at 37 °C in a humidified atmosphere with 5% CO_2_ for a period of 0–28 days. After each incubation period (1, 7, 14, 21, and 28 days), 200 µL of conditioned medium from CGF-permeated implants (I-CGF) was collected and replaced with 200 µL of fresh culture medium. Then, I-CGF was centrifuged at 1500 rpm for 10 min at room temperature, and the supernatant was collected and stored at −80 °C until analysis. The growth factors VEGF, TGF-β1, and BMP-2, and the matrix metalloproteinases MMP-9 and MMP-2 released in I-CGF, were analyzed using commercial human ELISA kits, according to the manufacturer’s instructions. The total quantity of soluble factors in the medium collected at all time points was quantified and reported as a mean value at each time point.

### 4.4. Endothelial Cell Adhesion on CGF-Permeated Implants

The human microvascular endothelial cells (HMEC-1), obtained from Dr. Thomas J. Lawley, were cultured as previously described [52]. To verify the biocompatibility of CGF-permeated implants, HMEC-1 cells (4 × 10^4^) were loaded with DilC18 vital fluorescent dye, as previously described [18], and seeded on the implants with/without CGF. After 16 h, HMEC-1 cells adherent to implants were monitored at fluorescence microscopy, images were captured by NIS-Elements F 3.0., and fluorescence intensity quantified by ImageJ.

### 4.5. Cell Culture and Osteogenic Differentiation

Undifferentiated human bone-marrow-derived mesenchymal stem cells (hBMSC, ATCC-PCS-500-012) were cultured in mesenchymal stem cell basal medium (BM, ATCC PCS500030) supplemented with 7% FBS, 100 IU/mL penicillin/streptomycin, 2.4 mM, 125 pg/mL FGF-b, and 15 ng/mL IGF-1 (ATCC, Milan, Italy), at a density of 5 × 10^3^ cells/cm^2^ and incubated for 24 h at 37 °C under 5% CO_2_. To induce osteogenic differentiation, hBMSC were cultured in osteogenic medium (OM), (DMEM with 10% FBS, 100 IU/mL penicillin/streptomycin, 2 mM L-glutamine (Corning, Manassas, VA, USA), 10 mM β-glycerophosphate (BGP), 100 nM dexamethasone, 100 μM ascorbic acid 2-phosphate (AA) (Sigma Chemical Co., Milan, Italy), or cultured in BM with the presence of implant plus CGF supplemented with 7% FBS, 100 IU/mL penicillin/streptomycin, 10 mM BGP, and 100 μM AA (Device) for 21 days. The medium was replaced at a rate of 50% every 3 days. In all experiments, the implant coated with CGF was placed into a sterile transwell insert (TC-inserts, Sarstedt, Nümbrecht Germany) with a semipermeable membrane at the bottom (pores of 0.4 μm) and inserted into the 12-well culture plates (an insert in each well).

### 4.6. Alizarin Red Staining

Alizarin red S (Sigma) staining (ARS) was carried out as described in [53]. Briefly alizarin red S stain 2% solution in distilled water, was adjusted to pH 4.2 by adding ammonium hydroxide drop by drop while stirring, using an electrode pH meter. The solution was then filtered through a 0.45 μm microfilter (Millipore Corporation, Bedford, MA, USA), and kept in an amber bottle. This solution was filtered through a 0.22 μm microfilter immediately before use. BMSC, 2 × 10^4^ viable cells/mL, were seeded in a 12-well plate. After 24 h, the culture medium was refreshed. Cells were grown in BM, OM, or CGF-permeated implant, for 21 days. ARS of BMSC cells was performed for 21 days to detect osteoblast calcification. Cells were washed twice with PBS, fixed in 4% (*v*/*v*) paraformaldehyde in PBS for 15 min, washed with distilled water three times, and then stained by alizarin red S staining solution. After being rinsed twice with distilled water, the cells were photographed.

### 4.7. Real-Time PCR

Total RNA was extracted from the cells grown in a 12-well plate using the Trizol (Sigma, Merck Life Science S.r.l., Milan, Italy) following the manufacturer’s protocol. The reverse transcriptase reaction (20 μL) was conducted using 1 μg of total RNA, random primers, and MultiScribe^®^ Reverse Transcriptase (Applied Biosystem, Monza, Italy) according to the manufacturer’s protocol. Quantitative gene expression analysis was performed in a CFX Connect Real-time System (Biorad, Segrate, Italy) using SYBR Green technology (FluoCycle-Euroclone, Milan, Italy). Primers used in real-time PCR are reported in Table 2. The efficiency of each primer was tested running a standard curve in duplicate. The quantification was performed using the ΔΔCT method and *Gapdh* gene was used as an internal control for normalization. Fold change in mRNA expression was relative to untreated BMSC (CTR). The specificity of PCR products was confirmed by melting curve analysis. The identity of the amplified products was confirmed by sequencing analysis.

### 4.8. Clinical Applications of CGF-Permeated Implants

For clinical applications of dental implants, 10 healthy subjects, aged between 60 and 72 years old, were enrolled to be rehabilitated with 20 implants (IML, Immediateload^®^, Swiss) of which 10 were CGF-permeated. Informed consent was obtained from all patients included in this study in accordance with the Declaration of Helsinki.

All subjects were appropriately examined before surgery and two radiographic examinations were required: (i) dental arches orthopantomography to have a global assessment of the patient, to detect sites of monoedentulism, or to plan multiple implant insertions in the absence of two or more dental elements; (ii) cone-beam computed tomography examination of the affected arch to study the thickness, height, and quality of the rehabilitated bone tissue.

All patients enrolled in the study had never undergone implant operations in their life. They had no congenital pathologies with a negative family history. None of them followed particular drug therapies, and they had no drug allergies.

The split-mouth design, which provides rehabilitating one half arch with traditional IML implants, and the other half arch with CGF-permeated implants in the same patient, was used.

Patients with poor oral hygiene, insufficient space available for the introduction of the fixture, bruxism, rehabilitations with angulation correction of over 25° with respect to the implant axis, or rehabilitations with excessive cantilever were excluded from the study.

The post-operative pain was evaluated through a questionnaire filled in 1 day and 7 days after surgery. Pain was specifically self-assessed daily through a 0–10 visual analogue scale (VAS), where 0 = no pain and 10 = the worst conceivable pain.

Moreover, the healing times, any problems on the implant prosthesis, the intraoral radiographic (bone resorption, any exposed implant threads), as well as the peri-implant probing (bleeding or pain on probing) up to 6 months after surgery were evaluated.

### 4.9. Limitations of the Study

The present study was innovative because, for the first time in humans, implants permeated with and without CGF were compared.

However, an important limitation is the fact of having to rely on exclusively clinical success and not being able to take a bone sample for histological examination.

Furthermore, our study refers to a limited number of subjects recruited and to a relatively short follow-up; in fact, the data present an evaluation of only 6 months after surgery. The operated patients will certainly be the subject of further studies to assess the success of CGF even in the long term.

### 4.10. Statistical Analysis

Values were expressed as mean ± SD for the number of experiments indicated in the legends to the Figures. Differences between the two groups were determined by unpaired Student’s *t*-test. Multiple comparisons were performed by analysis of variance (ANOVA). Fisher’s exact test was used for nominal data such as bleeding on probing. Statistically, a significant value was considered for *p* < 0.05. Data were analyzed using SPSS Statistics for Windows, version 17.0 (SPSS Inc., Chicago, IL, USA) software.

## Figures and Tables

**Figure 1 ijms-23-15164-f001:**
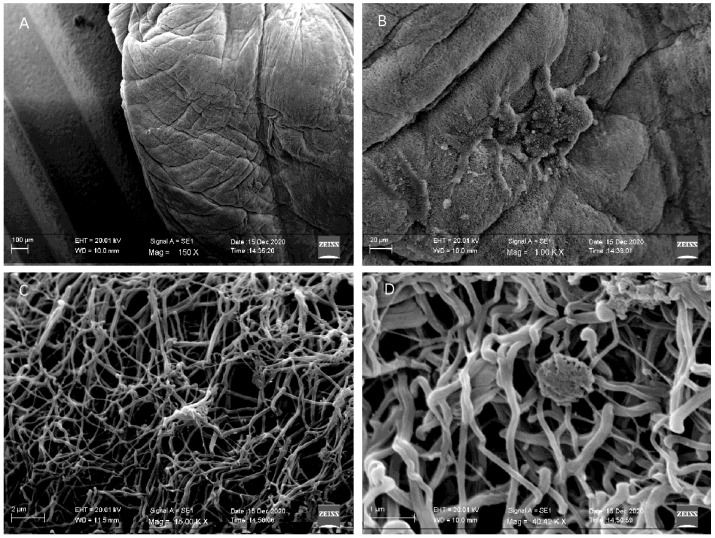
Scanning electron microscopy analysis of CGF-permeated implant. (**A**) CGF was able to adhere to the implant surface almost completely. (**B**,**C**) CGF fibrin forming a dense network conteining few corpuscular elements. (**D**) A corpuscular element entangled in the fibrin network.

**Figure 2 ijms-23-15164-f002:**
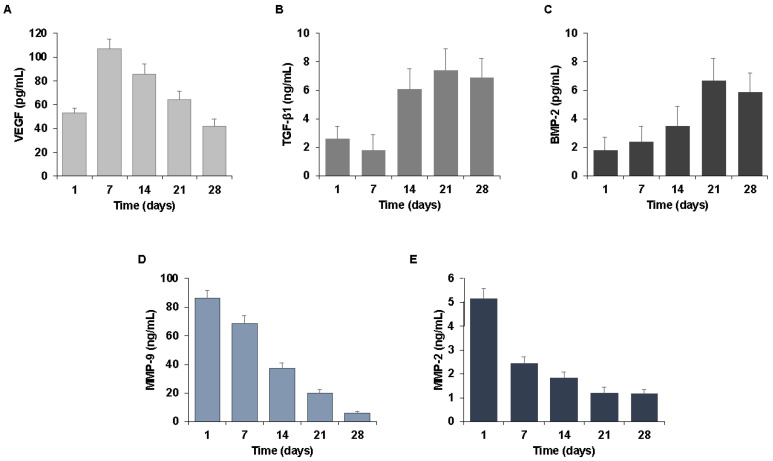
Growth factors and matrix metalloproteinases released by CGF-coated implants. Implants permeated with CGF were incubated in cell culture medium for a period of 0–28 days. At the indicated times (1, 7, 14, 21, and 28 days), the conditioned media were collected. The growth factors (**A**) VEGF, (**B**) TGF-β1, (**C**) BMP-2, the matrix metalloproteinases (**D**) MMP-9, and (**E**) MMP-2 were quantified by ELISA. The results are expressed as the means ± SD of triplicate measurements from three independent experiments.

**Figure 3 ijms-23-15164-f003:**
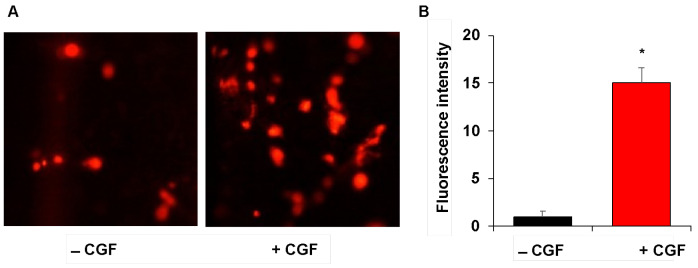
Endothelial cell adhesion on CGF-permeated implants. Implants without/with CGF (−/+ CGF) were incubated in the presence of DilC18-labeled endothelial cells and their adhesion was monitored by a fluorescence microscope (×100 magnification) (**A**). The fluorescence intensity was quantified by ImageJ software and reported as arbitrary units (**B**). The results are expressed as the means ± SD of triplicate measurements from three independent experiments. (* *p* < 0.05 versus implants without CGF).

**Figure 4 ijms-23-15164-f004:**
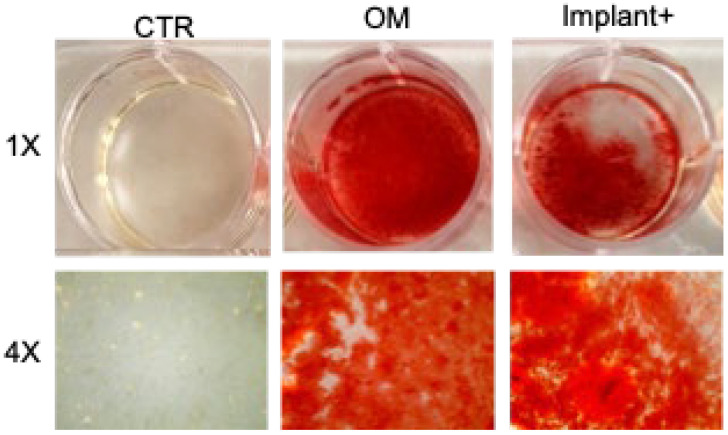
Osteogenic effect of the CGF-permeated implants. Alizarin Red staining of hBMSC cultured in basal medium (BM) (control, CTR), osteogenic medium (OM), and BM + BGP + AA + CGF-permeated implant (Implant+), for 21 days.

**Figure 5 ijms-23-15164-f005:**
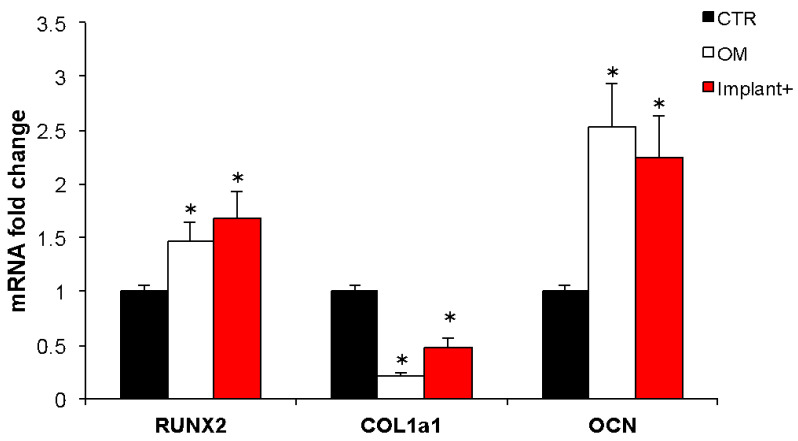
Effect of CGF-permeated implants on osteogenic gene expression. mRNA abundance of RUNX2, COL1a1, OCN in hBMSC cultured in basal medium (BM) (control, CTR), osteogenic medium (OM), BM + CGF-permeated implant (Implant+) for 21 days. *Gapdh* was used as a housekeeping gene for normalization. Fold change in mRNA expression was relative to CTR. The results were expressed as the means ± SD of triplicate measurements from three independent experiments (* *p* < 0.05 versus CTR).

**Figure 6 ijms-23-15164-f006:**
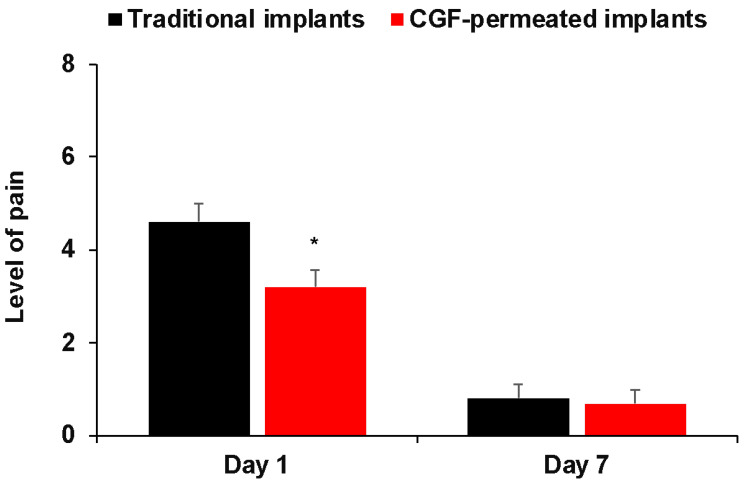
Post-operative level of pain between two groups (traditional implants and CGF-permeated implants) at different time points. * *p* < 0.05 versus traditional implants.

**Figure 7 ijms-23-15164-f007:**
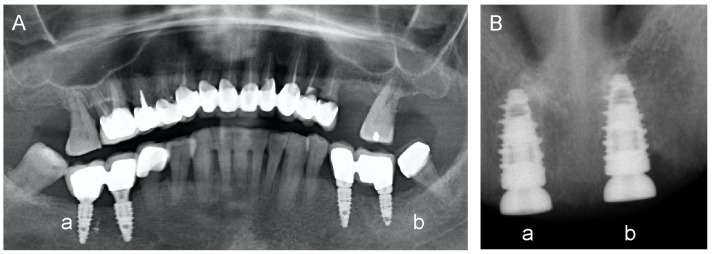
Sample images of dental implants with or without CGF permeation. Patients treated with split-mouth with universe type IML implants. (**A**) CGF-permeated implants in quadrant IV (a, right side) showed better crestal level and probing, and less bleeding when compared with those executed in quadrant III without CGF (b, left side), showing bone resorption problems. (**B**) CGF-permeated implants in quadrant I (a, right side) and in quadrant II without CGF (b, left side).

**Figure 8 ijms-23-15164-f008:**
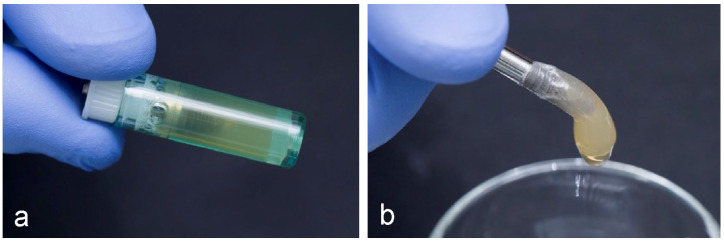
(**a**) CGF inserted into dedicated tube containing an Immediateload^®^, Swiss dental implants, diameter 4 mm, height 8 mm; (**b**) CGF-permeated implant after centrifugation using Round up (Silfradent srl).

**Table 1 ijms-23-15164-t001:** Post-operative course for traditional and CGF-permeated implants.

	Traditional Implants	CGF-Permeated Implants	*p* Value ^(1)^
N (%)	10 (50.0)	10 (50.0)	
Bleeding on probing			0.10
No	5 (50.0)	8 (80.0)	
Yes	5 (50.0)	2 (20.0)	
Successful implantation			0.35
No	2 (20.0)	1 (10.0)	
Yes	8 (80.0)	9 (90.0)	

^(1)^ Fisher’s exact test.

**Table 2 ijms-23-15164-t002:** Oligonucleotides used for real-time PCR analysis.

Gene Name	Accession Number	Sequences	pb
*RunX2*	NM_001278478.2	F: gacaaccgcaccatggtgg	160
		R: tctggtacctctccgaggg	
*Col1a1*	NM_000088.3	F: agggaatgcctggtgaacg	90
		R: gagagccatcagcacctttg	
*Ocn*	NM_199173.6	F: gctacctgtatcaatggct	111
		R: cgatgtggtcagccaactc	
*Gapdh*	AJ005371.1	F: atggccttccgtgtccccac	245
		R: acgcctgcttcaccaccttc	

*RunX2*, runt-related transcription factor 2; *Col1a1*, collagen type I α1; *Ocn,* osteocalcin; *Gapdh,* glyceraldehyde-3-phosphate dehydrogenase.

## Data Availability

The data presented in this study are available on request from the corresponding author. The data are not publicly available due to privacy.

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
