# Peer review of "Use of CGF in Oral and Implant Surgery: From Laboratory Evidence to Clinical Evaluation"

_ijms, 2022, doi:10.3390/ijms232315164_

Round 1
Reviewer 1 Report
This manuscript describes the osteogenic performance of CGF-permeated dental implants both in-vitro and in-vivo. This is an interesting topic, where readers could be inspired by the results presented that CGF-permeated titanium significantly improves the osseointegration properties. However, some major concerns are commented below:
Comments:
Many studies have already reported the in-vitro release profile of crucial factors from the CGF (e.g., VEGF). Moreover, the interaction between CGF and BMSCs was also extensively studied. From this perspective, the novelty of the study is weak, and the new findings lack. Additionally, this manuscript lacks a key in-depth discussion in which the present data should be compared with those obtained in a similar system rather than only a simple description and interpretation of them. The authors should also concisely structure the discussion section since there seem to be a lot of unnecessary descriptions that might be irrelevant to the current study.
If the abovementioned major concerns are all cleared by the authors, this paper could be re-evaluated.
Apart from major concern, some specific concerns are:
(1) Figure 1 – The SEM images that indicate the titanium implant surface permeated by CGF are poor. It will be confusing for the readers to simply provide the SEM images without a clear interpretation. For instance, there is no explanation for Figure 1A or Figure 1B. Please revise and indicate them both in the main text and in the images.
(2) Figure 4 – Please subdivide the figures into (A) (B) (C), as indicated in the figure caption. Additionally, the figure (1X CTR) should be improved.
Author Response
- Many studies have already reported the in-vitro release profile of crucial factors from the CGF (e.g., VEGF). Moreover, the interaction between CGF and BMSCs was also extensively studied. From this perspective, the novelty of the study is weak, and the new findings lack. Additionally, this manuscript lacks a key in-depth discussion in which the present data should be compared with those obtained in a similar system rather than only a simple description and interpretation of them. The authors should also concisely structure the discussion section since there seem to be a lot of unnecessary descriptions that might be irrelevant to the current study.
Response: We thank the reviewer for his comments and suggestions. We acknowledge that there are several studies on the release of CGF growth factors and the interaction between CGF and BMSCs. However, to the best of our knowledge, this is the first study characterizing the osseointegration properties of CGF-permeated titanium dental implants. In fact, in this work we do not study CGF as such but the titanium dental implant permeated with CGF. We have demonstrated that our innovative procedure is able to permeate the implant with CGF producing a biologically active and biocompatible surface capable of releasing growth factors, improving endothelial cell adhesion and inducing BMSC osteogenic differentiation, facilitating the osseointegration process and implant success in vivo.
Taking into account the reviewer's suggestions, in the revised manuscript we have clarified the novelties of our study (page 3, lines 110-113 lines; page 9, lines 299-303 lines) and deleted sentences from the discussion section that were not relevant to the current study (page 9, lines 332-338 lines; page 10, lines 350-354).
- Figure 1 – The SEM images that indicate the titanium implant surface permeated by CGF are poor. It will be confusing for the readers to simply provide the SEM images without a clear interpretation. For instance, there is no explanation for Figure 1A or Figure 1B. Please revise and indicate them both in the main text and in the images.
Response: According to the reviewer’s comment, we improved the explanations for Figure 1, by adding more details both in the main text (page 3, lines 120-125) and in the figure caption.
- Figure 4 – Please subdivide the figures into (A) (B) (C), as indicated in the figure caption. Additionally, the figure (1X CTR) should be improved.
Response: We decided to remove the subdivision of the figure into (A) (B) and (C) from the Figure 4 caption. Moreover, we noticed that there was an error in the uploading of figure 1X CTR, because there was only a white box. Now the panel shows the Alizarin Red staining of hBMSC cultured in Basal Medium (BM), demonstrating the absence of mineralized nodules in the control condition.
Reviewer 2 Report
Dear Authors,
thank you for the possibility to review this interesting paper. There are still some flaws and inaccuracies that should be explained or corrected when preparing the revised version of the manuscript:
- the title shoud contain words that it is a dental study (not medical) or dental implant study, to be even more speciffic
- when talking about the smile esthetics, please note the perception of smile by the Patient - how it is important and seen by the patient:
- the most commonly usted in dentistry are PRP and PRF, whith different modifications, which should be impremented in the introduction, see the references
* Pietruszka P, Chruścicka I, Duś-Ilnicka I, Paradowska-Stolarz A. PRP and PRF-Subgroups and Divisions When Used in Dentistry. J Pers Med. 2021 Sep 23;11(10):944. doi: 10.3390/jpm11100944. PMID: 34683085; PMCID: PMC8540475.
- please, add more detailed inclusion crytieria (did the patients have previous medical records, was the patient treated with implants before, did they suffer from any gongenital disease)
- In the discussion section it would be valid to mention the analysis of surfaces of implants, because that is what guarantees the healing:
* Hadzik J, Kubasiewicz-Ross P, Simka W, Gębarowski T, Barg E, Cieśla-Niechwiadowicz A, Trzcionka Szajna A, Szajna E, Gedrange T, Kozakiewicz M, Dominiak M, Jurczyszyn K. Fractal Dimension and Texture Analysis in the Assessment of Experimental Laser-Induced Periodic Surface Structures (LIPSS) Dental Implant Surface-In Vitro Study Preliminary Report. Materials (Basel). 2022 Apr 7;15(8):2713. doi: 10.3390/ma15082713. PMID: 35454406; PMCID: PMC9027964.
* Kubasiewicz-Ross P, Hadzik J, Gedrange T, Dominiak M, Jurczyszyn K, Pitułaj A, Nawrot-Hadzik I, Bortkiewicz O, Fleischer M. Antimicrobial Efficacy of Different Decontamination Methods as Tested on Dental Implants with Various Types of Surfaces. Med Sci Monit. 2020 Feb 20;26:e920513. doi: 10.12659/MSM.920513. PMID: 32078588;
- please, discuss the oral health condidtion in eldery people, especially in multi-national studies - this could show how many of those citisens would need also implant treatment, eg.:
* Skośkiewicz-Malinowska K, Noack B, Kaderali L, Malicka B, Lorenz K, Walczak K, Weber MT, Mendak-Ziółko M, Hoffmann T, Ziętek M, Walter M, Kaczmarek U, Hannig C, Radwan-Oczko M, Raedel M. Oral Health and Quality of Life in Old Age: A Cross-Sectional Pilot Project in Germany and Poland. Adv Clin Exp Med. 2016 Sep-Oct;25(5):951-959. doi: 10.17219/acem/63952.
- I woudl also suggest adding a point of coatings to healing of the bone around the dental implant, eg. chitosan coatings,
- Last but not least it would be valid to add the point of implant loading on the survival rate, also in the discussion:
* Krawiec M, Olchowy C, Kubasiewicz-Ross P, Hadzik J, Dominiak M. Role of implant loading time in the prevention of marginal bone loss after implant-supported restorations: A targeted review. Dent Med Probl. 2022;59(3):475–481. doi:10.17219/dmp/150111
- last point is that the paper should also contain the limitations of the study (as a separete chapter)
After those changes, the paper should be reevaluated. Thank you
Author Response
1) the title should contain words that it is a dental study (not medical) or dental implant study, to be even more specific
Response: We thank you for your valuable suggestion. We changed the title from: “Use of CGF in regenerative medicine: from laboratory evidence to clinical evaluation” to: “Use of CGF in oral and implant surgery: from laboratory evidence to clinical evaluation”.
2) when talking about the smile aesthetics, please note the perception of smile by the Patient - how it is important and seen by the patient:
Response: In the present work we have not talked about smile esthetics. In fact, we have compared the use of the dental implants permeated with CGF or not permeated in the field of tissue regeneration. The aesthetic aspect was not considered because it didn’t change.
3) the most commonly used in dentistry are PRP and PRF, with different modifications, which should be implemented in the introduction, see the references
* Pietruszka P, Chruścicka I, Duś-Ilnicka I, Paradowska-Stolarz A. PRP and PRF-Subgroups and Divisions When Used in Dentistry. J Pers Med. 2021 Sep 23;11(10):944. doi: 10.3390/jpm11100944. PMID: 34683085; PMCID: PMC8540475.
Response: According to the reviewer's comment, we have improved the introduction section with the description of the three generations of platelet derivatives and their classification. We added the following paragraph to page 2, lines 83-92.
“ Based on their characteristics and methods of preparation, platelet derivatives can be classified into three different generations. The first generation is platelet-rich plasma (PRP), which contains several growth factors involved in tissue repair, but the use of anticoagulants and bovine thrombin is required to induce fibrin polymerization [10]. Currently, the PRP has been divided into two subclasses: Pure platelet-rich plasma (P-PRP) and Leukocyte and platelet-rich plasma (L-PRP) [9].
The second generation consists of platelet rich fibrin (PRF) and can be classified into three subgroups: Pure platelet-rich fibrin (P-PRF), Leukocyte- and platelet-rich fibrin (L-PRF), Injectable PRF (I-PRF) [9]. For its preparation, blood samples are collected without the use of anticoagulants or biological agents [10].”
4) please, add more detailed inclusion criteria (did the patients have previous medical records, was the patient treated with implants before, did they suffer from any congenital disease)
Response: As you suggested, we have added more detailed inclusion criteria as follows:
“All patients enrolled in the study had never undergone implant operations in their life. They had no congenital pathologies with a negative family history. None of them followed particular drug therapies, and they had no drug allergies.” Page 14, lines 552-554.
5) In the discussion section it would be valid to mention the analysis of surfaces of implants, because that is what guarantees the healing:
Hadzik J, Kubasiewicz-Ross P, Simka W, Gębarowski T, Barg E, Cieśla-Niechwiadowicz A, Trzcionka Szajna A, Szajna E, Gedrange T, Kozakiewicz M, Dominiak M, Jurczyszyn K. Fractal Dimension and Texture Analysis in the Assessment of Experimental Laser-Induced Periodic Surface Structures (LIPSS) Dental Implant Surface-In Vitro Study Preliminary Report. Materials (Basel). 2022 Apr 7;15(8):2713. doi: 10.3390/ma15082713. PMID: 35454406; PMCID: PMC9027964.
Kubasiewicz-Ross P, Hadzik J, Gedrange T, Dominiak M, Jurczyszyn K, Pitułaj A, Nawrot-Hadzik I, Bortkiewicz O, Fleischer M. Antimicrobial Efficacy of Different Decontamination Methods as Tested on Dental Implants with Various Types of Surfaces. Med Sci Monit. 2020 Feb 20;26:e920513. doi: 10.12659/MSM.920513. PMID: 32078588;
Response: In the discussion section, we have introduced a paragraph regarding the implant surface as the reviewer advised. Moreover, we have referred to the two suggested references:
“Dental implant surfaces have now achieved outstanding performances. Common implant surfaces are classified into two great categories: smooth and treated. The implant micro-surface and nano-surface modifications have been proven to affect cellular responses such as cell adhesion, proliferation, differentiation, and migration, thus influencing bone healing. Due to surface modifications, it was possible to overcome the adverse effects of length reduction and the unfavorable crown–implant ratio of short implants, shortening the time needed to achieve secondary stability and deliver prosthetic restoration [35]. The decontamination of the implant surface is also of fundamental importance.This procedure is useful for prosthetic loading and to minimize peri-implantitis, which is the main cause of implant failure [36].” Page 9, lines 289-298.
6) please, discuss the oral health condition in eldery people, especially in multi-national studies - this could show how many of those citisens would need also implant treatment, eg.:
Skośkiewicz-Malinowska K, Noack B, Kaderali L, Malicka B, Lorenz K, Walczak K, Weber MT, Mendak-Ziółko M, Hoffmann T, Ziętek M, Walter M, Kaczmarek U, Hannig C, Radwan-Oczko M, Raedel M. Oral Health and Quality of Life in Old Age: A Cross-Sectional Pilot Project in Germany and Poland. Adv Clin Exp Med. 2016 Sep-Oct;25(5):951-959. doi: 10.17219/acem/63952.
Response: Thanks to the reviewer for the suggestion. We have added the following paragraph in the text:
“The importance of correct implant rehabilitation is also fundamental for improving one's real and perceived quality of life. Numerous studies show that this phenomenon is particularly evident in vulnerable groups such as the elderly [51]. Most of the older adults perceived that the most affected dimensions were psychological discomfort and functional limitation. Age, educational level, marital status, type of insurance, and level of income had a statistically significant relationship with oral health-related quality of life.” Page 11, lines 422-427.
7) I would also suggest adding a point of coatings to healing of the bone around the dental implant, eg. chitosan coatings.
Response: Thanks to the reviewer for the suggestion, we will consider the addiction of a chitosan coating around the dental implant in future studies.
8) Last but not least it would be valid to add the point of implant loading on the survival rate, also in the discussion:
* Krawiec M, Olchowy C, Kubasiewicz-Ross P, Hadzik J, Dominiak M. Role of implant loading time in the prevention of marginal bone loss after implant-supported restorations: A targeted review. Dent Med Probl. 2022;59(3):475–481. doi:10.17219/dmp/150111
Response: According to the reviewer's comment, we added the following paragraph:
“We had excellent results in all patients treated with both immediate loading and conventional (late) loading. Immediate loading provides benefits such as short treatment time, the elimination of the second surgery required for later loading protocols, the protection of the gingival papilla, an immediate esthetic effect, and high patient satisfaction [50]. However, no significant differences in the survival rate were reported between the immediately and conventionally loaded implants.” Page 11, lines 416-421.
9) last point is that the paper should also contain the limitations of the study (as a separate chapter)
Response: Thanks to the reviewer for the suggestion, we added at the end of the paper the following paragraph:
“The present study was innovative because, for the first time in humans, implants permeated with and without CGF were compared.
However, an important limitation is the fact of having to rely on exclusively clinical success and not being able to take a bone sample for histological examination.
Furthermore, our study refers to a limited number of subjects recruited and to a relatively short follow-up, in fact the data present an evaluation of only 6 months after surgery. The operated patients will certainly be subject of further studies to assess the success of CGF even in the long term.” Page 14, lines 570-577.
Round 2
Reviewer 1 Report
Dear authors,
Thank you for the revision. However, the manuscript still lacks novelty, even for evaluating the osteogenicity of the CGF-permeated titanium surface has been extensively studied.
Unfortunately, the quality of the work does not meet the standard of the IJMS journal; therefore, I recommend rejecting the manuscript.
Reviewer 2 Report
Dear Authors,
thank you for the corrections and explanations. I have nothing more to add. Thank you